# Lessons from Building a Sustainable Healthcare Exchange between the Netherlands and Cuba

**DOI:** 10.3390/ijerph191811742

**Published:** 2022-09-17

**Authors:** Paul Jonas, Eduardo Garbey Savigne, Mark Koster, Imti Choonara

**Affiliations:** 1Department of Public Health and Primary Care, Leiden University Medical Center, 2333 ZA Leiden, The Netherlands; 2International Relations Office, University of Medical Sciences of Havana, Havana 11600, Cuba; 3Department of Primary Care, Vrije Universiteit-Amsterdam Medical Center, 1081 HV Amsterdam, The Netherlands; 4Department of Child Health, Derby Medical School, University of Nottingham, Derby DE22 3DT, UK

**Keywords:** international, medical, exchange, Cuba, public health practice, Netherlands, primary care, prevention

## Abstract

Over the past ten years, seven Dutch Universities have built a sustainable exchange with seven institutes in Cuban healthcare. The exchange was initiated by the Leiden University Medical Centre and the University of Medical Sciences of Havana. Cooperation with Cuba was chosen as Cuba has excellent primary healthcare and has a strong focus on prevention and public health. These were considered important due to the major contribution of non-communicable diseases to morbidity and mortality in the Netherlands. Exchanges have occurred with Dutch health professionals and students visiting Cuban healthcare institutions and Cuban postgraduate students studying in the Netherlands. There has been an increased awareness of the importance of public health and prevention in Dutch professional organizations following the exchange. The exchange has also helped to break the scientific and economic US blockade of Cuba and resulted in joint publications. In this review we described the process, key aspects, results and lessons learned in this process. Collaboration between Cuba (a middle income) and the Netherlands (a high-income country) is possible.

## 1. Introduction

Health-/cost-benefit relationships in our mainly curative healthcare systems are moving beyond their break-even points [1]. On top of this, non-communicable diseases (NCDs) such as ischaemic heart disease, malignancies and accidents are increasing all over the world. Therefore, many governments want to put a greater emphasis on public health and prevention. The COVID-19 pandemic has convinced many people of the importance of both public health and disease prevention. In the Netherlands, NCDs are the major cause of morbidity and mortality. In 2020, NCD mortality was significantly greater than COVID-19 mortality [2]. A focus on public health and prevention is therefore important. In the Netherlands, practical experience in the area of prevention in the community is still limited. There is therefore an important incentive to learn more from a country with a lot of experience in that area, i.e., Cuba.

Cuba also has a strong motivation for exchange with a European country. Cuban science and the economy suffer from an economic and financial blockade by the government of the United States, which most countries in the world have strongly rejected [3]. Alongside the will to cooperate, it is important to highlight the Cuban healthcare system. Despite the blockade and being a middle-income country, Cuba delivers good health outcomes—life expectancy is about 79 years and infant mortality is 4 per 1000 live births–which are comparable to the Netherlands [4].

There is a positive need for exchange from both Cuba and the Netherlands. The goals of the exchange were defined as scientific cooperation and understanding the Cuban approach to healthcare, in particular, their focus on prevention and public health. The existing contact between a staff member of the department of Public Health and Primary Care (PHEG, The Netherlands) in Leiden University Medical Center (LUMC) with Universidad de Ciencias Medicas de la Habana (UCMH, Havana, Cuba) was the starting point for developing a joint program and, on 11 July 2012 in Havana and 12 May 2013 in Leiden, the Rector of UCMH and the Dean of LUMC signed a Memorandum of Understanding. This paper describes the exchange and the benefits and challenges.

## 2. Why Cuba?

The reasons for going to Cuba need some explanation. Since 1970, Cuba has made prevention, primary- and community-care the cornerstone of its healthcare system. This has been visible when comparing the impact of COVID-19 with nearby countries [5]. Cuba’s primary care system is focused on the family doctor, who usually will live within the community. The doctor works from the “consultorio” in conjunction with at least one nurse. Working as a team, they ensure every person is seen twice each year [6]. This includes house calls to see patients in their home. This systematic approach allows them to know each person medically and socially. This includes knowing the individual within their family setting. This approach means that the primary care team is aware of families with social problems and greater attention can be paid to these families. This allows the earlier identification and also prevention of child abuse and neglect, as well as domestic violence. Each primary care team is responsible for up to 300 families. Assuming there to be 4 individuals in a family, the primary care team is responsible for approximately 1200 people. All Dutch residents are registered with a local general practice of their own choice. One full-time GP (General Practitioner) provides ongoing medical care to an average of 2350 patients, with both male and female patients in all age groups. The GP is the gatekeeper to hospital and specialist care. They offer out-of-hour services by GP cooperatives across the whole country.

Most Dutch GPs work in small practices (two to five practitioners) located close to the community. Most practices are owned by the GPs themselves. Over the last decade, collaboration among practices has increased, moving towards larger teams and organizational networks. They have included other disciplines, such as physical therapists, psychologists and community nurses.

The success of Cuban investment in primary care is illustrated by their reduction in the number of low-birth-weight infants [7]. Low-birth-weight infants are at greater risk of both morbidity and mortality. Socioeconomic inequalities and poor nutrition (both malnutrition and obesity) are key factors in pregnant women predisposing to low-birth-weight infants. By focussing on both nutrition and ensuring adequate social welfare, Cuba has managed to reduce the number of low-birth-weight infants. This has been a major contributory factor to the low child mortality rates in Cuba.

Another example of the approach of Cuba to prevention is the public health measures in relation to natural disasters, mainly hurricanes [8]. Hurricanes, unfortunately, are a regular occurrence in Cuba. With climate change, they are likely to increase. Cuba educates children in school about hurricanes. Because hurricanes can be predicted by meteorological offices, the population of Cuba are warned in advance of any impending hurricanes. Where necessary, towns and cities are evacuated, and people housed in emergency centres. Following the hurricane, health care teams visit vulnerable individuals at home. This approach has resulted in Cuba having fewer deaths following hurricanes than neighbouring countries in the Caribbean and the United States of America. Cuba also has a strong emphasis on education, thus making it an excellent location for Dutch medical students, nurses, GP-residents and other medical professionals to study. Education at all levels in Cuba is free and this has resulted in Cuba training many doctors. Cuba now has one of the highest doctor-to-population ratios in the world (one doctor for every one hundred and eight people). In contrast, the Netherlands has one doctor per two hundred and forty-five people. As well as training Cuban doctors, Cuba has established the Latin American School of Medicine (ELAM). ELAM trains doctors from around the world. Unlike many medical schools in the Global North, ELAM trains doctors from disadvantaged communities for free, on the basis that they will return to their communities to help improve health outcomes there. Many students are funded by NGOs (Non-Governmental Organizations) in their host country [9].

From the description above, the motivation for exchange with Cuba is clear. Cuba has an integrated healthcare system, built on systematic and active prevention inside and outside the medical setting. There is an active participation of neighbourhood residents in health prevention projects. Furthermore, Cuba has a medical education system that is rooted in public health and (international) solidarity.

## 3. Capacity Building Quantified

On 7 July 2012, a group of five doctors and the Policy Adviser on International Affairs of Leiden UMC (Leiden, The Netherlands), arrived at José Martí airport near Havana to explore cooperation with Cuba. In that visit to Havana, the Cuban and Dutch colleagues identified initial possible areas of cooperation. The opportunities identified were both educational as well as scientific: The former consisted of rotations of Dutch medical students/residents/professionals to Havana and lectures featuring Cuban healthcare in Holland. The latter included pediatric audiologic screening, the use of nanoparticles in cancer therapy and the possible introduction of the Cuban medicines Heberprot and Cimavax in Holland.

Educational capacity building, the Dutch exchange goal of main interest, started through rotations from Leiden UMC to Havana, Cuba in 2013. One GP lecturer (1 week) 2 physicians (4 weeks) and a medical Masters student all visited Cuba and gained valuable experience. Enhanced by student and staff publications and lectures, capacity building developed from 2014 in a regular manner. This started with individual rotations and in-depth courses (called “minors”) from LUMC.

Since 2016, other European universities and organizations have showed an interest in exchanges. In 2016, the universities VU/AMC of Amsterdam, Groningen and Ghent (Belgium) joined. In 2017, universities from Utrecht, Rotterdam and Witten-Herdecke (Germany) became involved. In 2018, the largest Dutch nurse organization (“Buurtzorg”) and, in 2021, the Radboud University of Nijmegen became involved. (Table 1). Before every extension with a new University, the Leiden University, as initiators of the Cuba exchange, shared vital information in a single transfer session. Leiden also offered support in program development and provided information by issuing an information bulletin twice a year. The interuniversity “Focusgroup Cuba” (see Section 4.2) played a central role on the Dutch side.

Aligned with capacity building, the content development of programs in Holland also followed a regular path, building on former Cuban experience and didactic working methods. Individual rotations started in 2013. In-depth courses (called “minors”) of third-year medical students started in 2014 in Leiden and later also involved Rotterdam. Interprofessional Education groups of GP teachers, resident nurses and GPs/Geriatric GPs started with Leiden University. By 2015, they were also being organized by Amsterdam (VU and AMC). Since 2014, Leiden and a Dutch organization for occupational health have organized an annual study week in Havana.

The two exchange goals of main interest for Cuba were scientific cooperation and the awareness of the preventative Cuban health system in Holland, developed at different speeds. Some of the collaborations did not succeed (e.g., CIM, BioCubaFarma), whereas others have proved sustainable (UCMH, ENSAP, CIGB). Scientific cooperation is often a slow process, and it was so in this case. During the last 10 years, seven Cuban PhD students did a part of their PhD in Holland. Two of them obtained their PhD in Leiden in 2019. In 2017, a Dutch PhD student started research on Cuban Circulos de abuelos (Grandparents circles), which did not lead to publication. There was research on the role of carbohydrates in cooperation with the Finlay Institute, which did lead to a joint publication in 2020 [10]. Only one joint publication must be seen as a weakness, taking into account the fact that collaboration between the UK and Cuba has resulted in several publications highlighting the positive aspects of Cuba [11,12]. The other goal, i.e., awareness of the Cuban healthcare system in Holland, was successful, however. Numerous presentations at both small and big (inter)national congresses [13], publications in student magazines and also in (inter)national medical journals [14,15,16] have put Cuba, its healthcare system and public health and prevention on the Dutch academic map.

## 4. Key Factors for Sustainability

Besides the quantitative aspects, it is important to pay attention to four aspects that turned out to be key factors for sustainability.

### 4.1. Central Organization

In Cuba, the coordinator of International Affairs had central overall coordination during the last ten years. Cuba has extensive experience with medical students and professionals coming from all over the world to do their internships or working visits. A well-organized international office with a pre-and post-doc section was established in Havana. Additionally, a Cuban doctor was assigned to every Dutch individual or group.

In the Netherlands, there was another dynamic in central organization. The exchange in Leiden started with two doctors supported by the Leiden International Office. In 2015, when more universities showed their interest, an interuniversity cooperation group of students and doctors from different universities, called “Focusgroup Cuba”, was founded. The group met and organized on a regular and voluntary basis and provided information on social media. In consultation with their Cuban counterparts, a fixed protocol gradually took shape. As a result, within a few years, interest in the Dutch academic world increased. Due to personal circumstances and COVID lockdowns, the activity of the core group declined after 2019. However, Cuban exchange with different Dutch universities was sufficiently established. The remaining members of “Focusgroup Cuba” recently have transformed into the so called “Cuban-European Scientific and Health Exchange Group”. Its members are representatives of almost all Dutch universities and four others (one in each of Belgium, Germany, England and Italy). In Dutch universities, at first, one doctor was responsible for all facets of the exchange, but after diversification into individual “minor” and interprofessional groups, each kind of activity received its own coordinator.

### 4.2. Involving People

In Holland, active student involvement proved to be a key dynamic in organizational development. From the beginning of the exchange, all returning students were asked to join an informal peer-information network, to which almost 50% agreed. It was an important factor leading to “Focusgroup Cuba” as we saw in the previous section. Both in organization and capacity building, besides “active involvement”, “sharing” was a central issue and played a decisive role in its sustainability. Sharing information, results and programs with visiting staff of other universities became the norm. In addition, students were stimulated to do something to share their knowledge after their internship. Although this was actually intended to let them better internalize what was learned [17], it also turned out to be an important PR (Public Relations) asset. Some of the students held lectures, whilst others wrote articles. In 2017, some of them even designed a student information website [18]. Others produced or were interviewed in little films on social media and the internet [19].

### 4.3. Multilevel Feedback and Evaluation

Feedback also turned out to be essential for the sustainability of the Cuban–Dutch exchange. By making it partly mandatory and partly self-chosen, it was present in all levels and activities. Feedback allowed the adjusting of programs and assessing of the attractiveness of the exchange. After the first five internships, Leiden University sent an evaluation to their Cuban counterparts [20]. Cuba delivered very valuable feedback. The first students going to Cuba had only a basic understanding of Spanish. The Cubans requested more instruction in Spanish before visiting and this was introduced, resulting in the students getting a greater benefit from their visit. Thereafter, continuous feedback was established from every individual, group or working visit. This involved individual reflection through reports using Bateson’s theory [21], qualitative research, qualitative “GOES” evaluations and closing symposia by “minor” student groups, closing symposia and reports from evaluation sessions of Interprofessional Education (IPE) groups in Amsterdam and Leiden. It turned out to be a prerequisite for sustainability. This emphasis on feedback was less present in Cuban education than in their healthcare system, which is remarkable. It was discussed several times with the Cuban colleagues. The Cubans expressed surprise at the lack of clinical contact for Dutch students in year 1. The differences between the importance of public health and prevention in the two systems became apparent.

### 4.4. Pre-Departure Training

Pre-departure training [22,23] increasingly was seen as a major asset on the Dutch side of the exchange. It consisted of four 2 h sessions for individual students, three evening sessions plus a full week for “minors”, and four monthly days with assignments and lectures by participants of mixed Interprofessional Education (IPE) groups. The content of the sessions consisted of a Spanish language class taught by native Cuban colleagues and knowledge of Cuban people and culture explained by former rotation students or professors with experience in Cuba. The latter was sometimes even organized by the Cuban Embassy in the Hague. In addition, there was functional, personal and team training and training in qualitative scientific research techniques. From the “minor” groups, (scientific) reports were expected and exams were organized. In this way, the exchange was incorporated as much as possible in the curriculum itself, which promoted sustainability. Though students often complained about the intensive preparation and study requirements during and afterwards, the decision to give serious and sound scientific and didactic foundation to all involved in the exchange (individuals as well as groups) proved to be important. This may seem obvious but at the time, in Holland, global health internships had the image of a holiday [24]. In fact, things turned out very well, given the student nomination of the Cuba-minor Leiden as the best minor in 2019 and the invariably high rating scores in the so called “GOES qualitative student feedback”. Setting a thorough pre-departure training was an important lesson learned, as confirmed by research [25].

## 5. Did the Results Meet the Set Goals of the Exchange?

The main goals of the Cuban–Dutch exchange program for Cuba were attaining the content- and fund-related benefits of scientific cooperation and an awareness of the Cuban approach of healthcare in Holland. For the Netherlands, it was most important to acquire more knowledge of and focus on prevention and public health both in Dutch universities and Dutch society as a whole. Below, we give an overview of the concrete impacts and compare them with the set goals (Table 2).

### 5.1. Impact for Cuba

In terms of scientific cooperation, seven Cuban PhD students worked in Holland (two attained their PhD in Leiden); three research items were realized: one on diabetes/brown fat, and one on nanotechnology use in cancer and schistosomiasis therapy. Despite cooperation in scientific work being minimal, it was impactful. The development (now entering phase one) is in cooperation with CIGB-Havana of a promising COVID-nose-spray vaccine. There is the expectation of some useful properties: counteracting all variants, preventing spread, storable at room temperature and low cost. The optimism is based on the outcomes of animal testing in Holland and research on a comparable nose-spray vaccine elsewhere [26].

Awareness of the Cuban approach to healthcare in Holland: this was realized by: (1) an annual Cuba working visit by the Dutch Occupational Health organization Occure; (2) two Cuban teachers and a coordinator spending weeks in the Netherlands to give lectures in Rotterdam, Leiden and Utrecht; and (3) numerous presentations of their Cuban experience by Dutch staff, residents and students at national and international congresses [13]. Additionally, publications in both student magazines and medical journals [14,15,16] have put Cuba, its healthcare system and public health and prevention firmly on the Dutch academic map.

### 5.2. Impact for the Netherlands

More than 300 Dutch medical students, nurses, residents, teachers and doctors experienced healthcare in Cuba in rotations and working visits. Qualitative research showed that public health and prevention, mother and child health, elderly care, cross-cultural communication and reflection on ethical dilemmas were major areas of learning. Evaluations revealed an increased knowledge of public health and prevention skills and importantly a change of attitude. This involved not merely seeing the individual but seeing individuals as part of a community. Consequently, changes were made in the focus and organization of GP consultation. These included more empathy and proactivity in doctor’s consultations. Thinking was more directed to prevention and neighborhood-oriented issues. In some waiting rooms, graphs of the average blood-pressure in the district through the years appeared. District prevention projects inspired by Cuba were started. Many students changed future professional plans from being a hospital specialist to becoming a GP or Global Health doctor.

Most eye-catching was the switch of the Nurse Organization, named Buurtzorg (14,500 persons), from a purely curative setting to a more preventative approach. District nurses have organized collective prevention in their mostly underserved districts, starting with formation of groups of local residents. Initial activities have focused on group coherence, mutual aid and feeling at home, and from there developing other health enhancing activities. Currently, in 18 Dutch cities, this collective district prevention system has been introduced and has the potential of becoming a total new national Dutch prevention approach.

The exchange has also caused a change in Dutch educational programs. In 2015, Cuban experience formed part of the input in the prevention part of the review of the national GP curriculum. In Leiden and Rotterdam, new prevention modules were implemented in the GP curriculum, and currently Cuban experience forms part of the input in the review of the national nurse curriculum.

### 5.3. Impact for Both Partners

Mutual respect and understanding increased, reflected by an increase in cross-cultural communication skills, many longer-lasting contacts between Cuban and Dutch students and even two Cuban–Dutch marriages.

The recently founded “Cuban-European Scientific and Health Exchange Group” is a continuation of the former “Focusgroup Cuba”. Its members are representatives of almost all Dutch Universities and four others (Belgium, Germany, England and Italy). The group has already delivered some provisional results: the start of strengthening the UK exchange with Cuba and positive contacts in the field of blended learning and medical technology between the University of Santiago de Cuba and Erasmus University in Rotterdam.

Considering the volume of exchange traffic between Cuba and the Netherlands, one can see some imbalance in favor of the Netherlands. Different reasons might be responsible for that. The first could be caused by the nature of the kind of exchange. Student rotations simply are easier to organize than cooperation in science. Finance also could play a role. In Cuba (a medium-income country), the state will only invest if the result is directly useful for the people. In Holland (a high-income country), the choice to go for a rotation or working visit is mainly the decision of the student/professional her/himself. Funding is mainly provided by students/professional themselves, with only a minor contribution by the state.

## 6. Conclusions

Before starting an exchange, there should be an assessment of the objectives and motivation in both partners. The active support of management at all levels is needed. A key aspect is the participation of people who feel a personal and substantive drive to expand the exchange.

There are four key factors for sustainability: (1) central organization is needed to ensure the exchange runs smoothly. (2) People need to be fully involved, i.e., given the opportunity to highlight problems and give suggestions for improving the exchange. Active student involvement strengthened the organization and increased the sustainability. (3) Multilevel feedback and evaluation are essential as circumstances are different for different groups and changes may occur with time in both countries. Feedback has been positive and has helped improve the quality of the exchange. (4) A thorough and extensive pre-departure training is required. We want to stress that speaking Spanish turned out to be a “conditio sine qua non”. Building the exchange into the regular curriculum is an important asset for sustainability and of course for student motivation. Active student involvement both strengthens the organization and increases sustainability.

Collaboration between a high- and middle-income country is both possible and desirable. Both Cuba and the Netherlands have benefitted from the exchange. The Dutch have had the opportunity to see a preventative health system in practice and bring ideas from Cuba to the Netherlands. The Cubans have had the opportunity to do some of their postgraduate studies in the Netherlands, seeing a different culture and also greater facilities in academic institutions. Most importantly, links have been established at both an institutional and a personal level. This will hopefully allow collaborative research between Cuba and the Netherlands.

## Figures and Tables

**Table 1 ijerph-19-11742-t001:** Cuban–Dutch Medical Educational and Scientific Capacity Building, 2012–2022.

Organisation	First Year of Joining Exchange	Content
Havana Medical University	2012	Rotations individual working visits, minor groups, (interprofessional) groups
Leiden University Medical Center	2012	Rotations individual working visits, minor groups, (interprofessional) groups
Havana Center of Molecular Imunologics (CIM)	2013	Research diabetes, brown fat
Havana Biogenetic Center (CIGB)	2013	Research cancer, nanotechnology COVID-19 nasal vaccine
Occure Occupational Health	2013	Annual working visit
Finlay Institute	2014	Schistosomatiasis-carbohydrate gold nanoparticle research
National Centre of Medical Genetics	2015	Incidental visits, groups, individuals
Universities of Amsterdam (VU/AMC), Groningen, Ghent(B)	2016	Rotations, individual, residents/lecturers groups
Havana Medical University	2016	First PhD student to Leiden (6 more in next years) + Surgeon
Universities of Rotterdam, Utrecht, Witten-Herdecke (Ger)	2017	Individual rotations, minor groups, working visits
Leiden Academy Vitality + Aging	2017	Research Circulos de Abuelos
National School of Public Health (ENSAP)	2018	Rotations, interprofessional groups (during one month)
Buurtzorg (largest Dutch Nurse Organisation)	2018	Nurses working visits
Bio Cuba Farma	2018	Incidental visits by students/to Leiden
Lecturers Havana University	2018	Three working visits, Leiden, Rotterdam
National Institute for Hygiene, Epidemiology and Microbiology	2018	Incidental scientific rotations
University of Nijmegen	2021	Individual rotations
Buurtzorg-ENSAP	2022	Sustainable Nurse Exchange program

**Table 2 ijerph-19-11742-t002:** Achievements of the exchange.

Visits to health institutions in Cuba by Dutch health professionals and students
Greater awareness of healthcare in Cuba by Dutch organizations
Postgraduate experience for Cuban students in Holland
Publications (a limited number)
Introduction of prevention programs in Holland
Major changes in Dutch medical curricula
Contact between Cuban and Dutch researchers

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
