# Peer review of "Lessons from Building a Sustainable Healthcare Exchange between the Netherlands and Cuba"

_ijerph, 2022, doi:10.3390/ijerph191811742_

Round 1

Reviewer 1 Report (Previous Reviewer 2)

General remarks

The manuscript corresponds to a resubmission of another one, entitled “Building Sustainable Healthcare Exchange between the Netherlands and Cuba” (ijerph-1752833-peer-review-v1), which was a review and is now a case report.

Specific remarks

In the review report on that first submission, I drew attention, above all, to formal aspects that (easily) could be improved. In fact, almost all of them were accommodated by the authors -- although the cover letter format and the lack of track changes in the manuscript are not the most convenient ones -- but, unfortunately, flaws of that kind still exist, which are all the more unacceptable the higher the number of authors.

To be clear, how is it possible that lines 154-155 are empty, that lines 227-229 are made up of (only) two final periods, and that some paragraphs are left-aligned? I apologize if I offend the authors, but this kind of flaw, which is so easily avoidable, leaves reviewers with the idea that the authors did not commit themselves, as they should, to the submission of a manuscript for publication, which is obviously a serious matter.

Author Response

Reviewer 2 Report (New Reviewer)

Round 2

Reviewer 2 Report (New Reviewer)

Thank you

This manuscript is a resubmission of an earlier submission. The following is a list of the peer review reports and author responses from that submission.

Round 1

Reviewer 1 Report

See attached file

Author Response

Dear Reviewer Please see the attachment.

Paul Jonas.

Reviewer 2 Report

General remarks

The manuscript presents the relationship of cooperation, in the health care sector, that has existed between Cuba and the Netherlands.

 Specific remarks

First of all, it is important to acknowledge that the manuscript is a review, and it is, obviously, as so that I will analyze it.

Being a review, obviously neither a great formalization nor a great extension is required. Even so, a minimum of rigor and adherence to what is standard is required. Let me be clear, as follows:

·       The use of bold, such as, for example, "sustainable exchange", in the abstract, is questionable;

·    The use of underlined words, such as, for example, “year” (page 3), is also questionable;

·       Frankly, I can't understand what "All fields" means in the Keywords list;

·       Even in visual terms, one can see that the manuscript has a lot of double blank spaces (between words), which may lead the reader to believe that the text was written in a hurry;

·       There are some typos, which must, of course, be eliminated;

·       Some sentences do not end with a period, others with a blank space before the period, and others with periods;

·       Some references appear after the period, when they should appear before;

·       Some claims are questionable, but I'm willing to give the benefit of the doubt. For example:

o   “The Covid-19 pandemic has convinced people of the importance of both public health and disease prevention.” (page 1). In my opinion, we are far from having obtained this conviction.

o   “In the Netherlands, solid practical experience in the area of prevention in the community still is scattered and minimal.” (page 1). In my opinion, this claim is too pessimistic.

Author Response

Dear Reviewer 2,

kind regards,

Paul Jonas.

Reviewer 3 Report

From my point of view, this paper is not a scientific study in its full sense.

It is a purely descriptive story, without and theoretical and/or empirical analysis, and without conclusions.

Author Response

Dear Reviewer,

Thank you for your comments with which we agree to a great extent.  In our extensively revised version we made some major adjustments (structure, chapter layout, logic, framing, conclusions)  and thus hope to meet your comments.

with kind regards,

Paul Jonas. 

Round 2

Reviewer 3 Report

The new version of the paper is better than the previous one but does not answer to my key concerns.

Thus, in my opinion, the paper should not be accepted.